# Male Fertility and Fatherhood in Chronic Myeloid Leukemia: Current Understanding and Future Perspectives

**DOI:** 10.3390/cancers16040791

**Published:** 2024-02-15

**Authors:** Ahmed Adel Elsabagh, Maria Benkhadra, Ibrahim Elmakaty, Abdelrahman Elsayed, Basant Elsayed, Mohamed Elmarasi, Mohammad Abutineh, Nabeel Mohammad Qasem, Elrazi Ali, Mohamed Yassin

**Affiliations:** 1College of Medicine, Qatar University, Doha P.O. Box 2713, Qatar; ie1703006@student.qu.edu.qa (I.E.); be1802020@qu.edu.qa (B.E.); me1803937@student.qu.edu.qa (M.E.); 2National Center for Cancer Care & Research, Hamad General Hospital, Doha P.O. Box 3050, Qatar; mariabk1997@gmail.com; 3Hematology Section, Medical Oncology, National Center for Cancer Care and Research (NCCCR), Hamad Medical Corporation, Doha P.O. Box 3050, Qatarnqasem@hamad.qa (N.M.Q.); 4One Brooklyn Health, Interfaith Medical Center, Brooklyn, NY 11213, USA; eali@interfaithmedical.org

**Keywords:** CML, leukemia, male fertility, TKI, transplant, interferon, bone marrow transplantation

## Abstract

**Simple Summary:**

In this review, the authors collected and demonstrated evidence concerning the effect of chronic myeloid leukemia (CML) and its treatment options on fertility in male patients. This became increasingly important as we began observing this disease in younger adults. We provide an overview and draw conclusions regarding different medications and treatment options and how these may affect the ability to conceive. This information is crucial for healthcare workers and patients, as it must be disclosed to patients when starting them on one of these agents.

**Abstract:**

Chronic myeloid leukemia (CML), while traditionally a disease of the elderly, has recently risen in incidence among younger patients. Hence, fertility concerns have emerged considering the disease process and treatments, especially with the current scarce and conflicting recommendations. This review explores the impact of CML treatments including the first-line tyrosine kinase inhibitors (TKIs) and other treatments on male fertility in chronic myeloid leukemia (CML) patients. The aim of this review was to compile the available evidence on male fertility to ultimately tailor treatment plans for male CML patients for whom fertility and future chances for conception pose a concern. The data available on the conventional and newer TKIs to address fertility concerns were reviewed, particularly the potential long- and short-term effects. Also, the possible side effects on subsequent generations were a crucial focus point of this review to reach a more comprehensive CML management approach. We found and compared the evidence on TKIs approved to treat CML. We also reported the effects of hydroxyurea, interferon, and transplantation, which are considered second-line treatments. Our findings suggest that these drugs might have an undiscovered effect on fertility. More research with larger sample sizes and longer follow-up periods is essential to solidify our understanding of these effects.

## 1. Introduction

Chronic myeloid leukemia (CML) is mostly a disease of older age groups; however, many cases have been encountered among younger adults and adolescents. It is also more common among males compared to females, with a ratio of 1.16:1 [1,2]. The incidence of CML is 1–2 cases per 100,000 adults depending on the age of the population and it makes up 15% of newly diagnosed leukemia in adults [3]. In countries with a young population, the median age at diagnosis is less than 50 years [4]. Abdullah et al. studied CML in adolescents and young adults and described that these patients had better prognosis according to Sokal and Euro scores. However, they also found that this category of patients usually presents with high white blood cell (WBC) counts and lower hemoglobin levels [5]. CML is caused by the Philadelphia chromosome, a translocation mutation between chromosomes 9 and 22 (t(9,22)), which creates fusion between BCR and ABL1, resulting in BCR::ABL. Most patients are diagnosed with CML during its chronic phase. This review aims to add male fertility to the equation to facilitate therapy individualization for younger male patients seeking to preserve their fertility during or after CML therapy, when applicable.

### CML Treatments

The treatment of CML has been revolutionized by tyrosine kinase inhibitors (TKIs), as they are able to directly target the Philadelphia chromosome, making it a treatable disease with more favorable outcomes. Patients can now expect a near normal life expectancy compared to the general population [6]. Before TKIs, interferon-a (IFN-a) and stem cell transplantation were the mainstay for CML treatment, but were associated with intolerable side effects. TKIs are now the first-line therapy for patients with CML and now include three generations [7]. Second-generation TKIs are usually used in high-risk patients and third-generation drugs are approved for resistant types of CML, especially those with the T3151 mutation [8,9]. These drugs have become essential due to the rise in drug resistance due to mutations in the BCR-ABL kinase.

Despite their evident efficacy in CML treatment, TKIs have several adverse effects including cardiovascular toxicities, which are especially high in patients with comorbid conditions. Hence, it is recommended that the ABCDE approach be used, which consists of the assessment of risk, antiplatelet medications, blood pressure, cholesterol, cigarette cessation, diet, diabetes, and exercise to prevent these potential associated adverse effects [10]. Specifically, nilotinib and ponatinib have demonstrated the most vascular-occlusive adverse events, while dasatinib is associated with pulmonary arterial hypertension. Imatinib has the best safety profile. A system where persistent monitoring, sufficient and timely intervention, and follow-up might be effective in minimizing these effects [11].

Even though second-generation TKIs (2G) have not shown superiority to imatinib in terms of overall survival when given as a first-line therapy, these drugs are preferred for younger patients. This is particularly applicable for female patients, since 2G TKIs provide a deeper, more rapid molecular response and higher rates of complete remission compared to imatinib, thereby allowing for the discontinuation of TKIs, which is crucial for fertility [12]. Numerous studies have been conducted on both animal models and humans to assess the effects of TKIs on fertility. These studies yield contradicting results, especially in terms of male fertility [13]. Four drugs are now used as first-line treatments for CML, including imatinib, bosutinib, dasatinib, and nilotinib. Unlike cytotoxic therapies that target the cell cycle, TKIs have a better profile related to preserving fertility. Hence, they are not considered a fertility risk to men. Nevertheless, studies show that there might still be an association between TKIs and impaired fertility, as they influence several tyrosine kinases that are important for spermatogenesis and sperm motility [14].

In this review, an outlook on the evidence relating TKI usage to male fertility and the potential effects it might have will be provided, accounting for the complexity of treatment choice in CML patients. A patient’s comorbidities, age, disease risk, response to treatment and medication adverse events, and cost are all major factors to consider when choosing therapy for CML patients.

In several studies, TKI adverse events have been described as tolerable and rare. Mild to moderate adverse events often resolve spontaneously or by simple means. A brief interruption of the treatment is another strategy if other methods are unsuccessful. These include pleuropulmonary, cardiovascular, and gastrointestinal complications. The type of TKI chosen for treatment depends on the patient’s comorbidities, age, and the risk of the disease, which is determined by scoring systems like Sokal, EUTOS, and ELTS scoring systems [12,15]. Another obstacle that may affect the management of CML patients is the financial aspect. Nonadherence is a major issue, since TKIs may be unaffordable for many patients. This leads to patients being treated with less optimal treatments like hydroxyurea, which is usually transiently used after diagnosis is established, leading to more adverse effects [16].

## 2. Findings

The data regarding the effects of CML on male fertility are very limited compared to those on female fertility and pregnancy. As for sexual functioning, no data were found specifically relating to patients with CML and the effect of the disease or medication on sexual functioning. Nevertheless, priapism has been associated with CML as a rare complication that is usually seen at first presentation and may hinder sexual functioning. Ali et al. also found that the best treatment in these cases is to start treatment of CML, which would enhance the resolution of priapism [17]. The study demonstrated that half of the cases of priapism in patients with leukemia are due to CML. They also discuss the fact that it might be the first manifestation that a CML patient presents with. They found that after treatment, the rates of priapism tend to reduce significantly, which is the most effective way for treatment. Anemia is a proposed cause of priapism in these patients. A major side effect of priapism that may hinder sexual functioning is erectile dysfunction, which is a complication of priapism in some cases. Hence, the early treatment of priapism is crucial. Surgical options like shunts, aspiration, and irrigation offer a faster modality for management. This relationship needs further investigation to provide a better understanding of the underlying pathogenesis and its prevention [17].

### 2.1. Hydroxyurea

Hydroxyurea was discovered and implemented in CML therapy after it showed the capability to prolong survival. It was one of the first-line treatments with interferon-alpha and busulfan before the introduction of TKIs [18]. Now, it is used as a debulking therapy (pre-treatment phase) and usually no complications are observed due to its short use. Nevertheless, due to several factors like financial status, the availability of TKIs, and performance status, hydroxyurea may sometimes still be used for CML treatment in many patients. Hence, the effects on male fertility due to hydroxyurea is not well studied in CML patients. However, hydroxyurea’s effects are well established in cases of other diseases that require hydroxyurea therapy for longer periods. Hydroxyurea is proven to have brief cytotoxic effects on dividing cells, and, hence, may affect spermatogenesis [19]. To be more precise, in a research study carried out using transgenic sickle cell mice exposed to hydroxyurea, it was found that sperm mobility was affected and testosterone levels were reduced. The researchers also observed Leydig cell hyperplasia in these mouse models compared to the control mice, which could be an effect of reduced testosterone. They concluded that this could most likely be attributed to an increase in nitric oxide, which is a metabolite of hydroxyurea and acts as a suppressor of testosterone synthesis, which leads to all other observed effects [20]. Several cases of azoospermia and decreased sperm production after hydroxyurea therapy has also been reported in the literature. These effects have also been shown to be reversible after cessation [21,22]. Consequently, the close monitoring of sperm count and morphology may be advised in clinical management when using hydroxyurea therapy, which might not be highlighted in current guidelines [23].

### 2.2. Interferon

Interferon-alpha (IFN-α) was also once the standard therapy for CML. It has the ability to target CML stem cells and induce remission in patients in the chronic phase of the disease and was associated with prolonged survival. IFN-α works on reducing pro-angiogenic vascular epidermal growth factor (VEGF), which is used as a biomarker that indicated CML severity [24]. Since it has been shown in many trials that TKIs alone do not completely destroy CML stem cells, there has been a rise in trials that use TKI and interferon-a combination therapy. This was shown to reduce the time needed to achieve remission than using a TKI alone. In addition, the increasing resistance to TKIs made it important to further study alternative therapies like IFN-α. Resistance is caused because the exposure of leukemic stem cells to many TKIs may lead to the failure of apoptosis. Combination therapy showed higher rates of complete remission in many studies [25,26]. However, it should be noted that in other studies, both imatinib alone and imatinib with IFN-α therapy showed similar progression-free survival rates [27]. Some of the undesirable effects of IFN-α include thrombocytopenia by inhibition of the proliferation of megakaryocyte progenitor cells [28]. The effect of IFN-α on male fertility has not been well documented in the literature. IFN-α has been used as a TKI alternative during pregnancy and showed less teratogenic effects [29]. In addition, it has also been used as an alternative in a patient who stayed on IFN-α during ovarian stimulation for fertility preservation, which was then successful, which could indicate significantly fewer adverse effects on fertility compared to TKIs [30].

### 2.3. TKIs

TKIs have also been found to influence almost all endocrine glands. This would lead to adrenal insufficiency, hypogonadism, and fertility impairment in both genders. Hormone deficiencies are typically managed with replacement therapy of reduced hormones [31]. Table 1 summarizes these findings and provides an overview of the drugs currently available as treatment options for CML and the relevant evidence regarding their effects on male fertility. Some studies found no effect of CML on male fertility. Gazdaru et al. published a case report concerning a case of CML resistant to TKIs. In this case, TKIs were replaced by interferon-a (IFN-a) in order to safely undergo ovarian stimulation for zygote cryopreservation. They successfully generated nine zygotes after preserving 12 oocytes. The patient later needed a hematopoietic stem cell transplantation (HSCT). This shows that fertility preservation in cases of gonadotoxic therapies and CML may be essential in some cases [30].

Tyrosine kinase inhibitors have been shown to have significant adverse effects on both female and male infertility. It affects sperm maturation and gonadal functions; however, studies that investigate the long-term effects after the discontinuation of medications are lacking. In terms of conception, male patients have no limitations to using TKIs while trying to conceive with partners and contraception is not encouraged, unlike with female patients. Nevertheless, it has been documented that TKIs affect c-Kit and PDGFR, which potentially diminish spermatogenesis and testosterone concentrations, which, in turn, decrease male fertility. In addition, several testicular cell types including Leydig cell precursors, adult Leydig cells, and gonocytes express PDGFR. Moreover, PDGF was shown to play a role in myeloid cell development, which helps sperm move through seminiferous tubules. PDGFR is usually expressed in Leydig cells in male testes as it plays a role in their development during gonadogenesis at the fetal period. c-Kit also has a role in sperm migration and survival [44]. As shown in Figure 1, c-Kit in mice was shown to play a role in spermatogenesis by affecting stem cells through their microenvironment, resulting in spermatogenesis and the differentiation of stem cells as well as the multiplication of stem cells. Kit-negative cells have the ability to either multiply by creating a Kit-positive spermatogonial cell or transform into a Kit-positive differentiating spermatogonia that produces sperm cells. The disturbance of this process may hinder spermatogenesis and, hence, male fertility. It is interesting that since spermatogenesis and cancer formation have similarities and similar physiological features like cell proliferation and survival, many signaling pathways are common in both processes. They were also shown to play a major role in testicular tumors, as their overexpression may lead to fibrotic disorders and malignancies [45]. Hence, it has been shown that the usage of TKIs like imatinib before puberty may have adverse effects on animal as well as human subjects.

A 2012 study by Yassin et al. studied the effects of imatinib, dasatinib, and nilotinib in CML patients and found a significant decrease in sperm parameters including sperm count, volume, motility, and morphology after 4 months of treatment. The study also suggested that this might be caused by a negative effect on the pituitary gland as serum LH, FSH, and testosterone. The effect on spermatogenesis was much less pronounced in patients using dasatinib compared to patients using imatinib or nilotinib [46,47]. A 2023 study by Abu-Tineh et al. was conducted on male patients with CML in chronic or accelerated phases receiving imatinib, dasatinib, and nilotinib. They also studied the effect this had on the patients’ offspring. They concluded that therapy had no effect of fertility, and their children were healthy with no increased risks of congenital abnormalities, impaired growth, or malignancies [48]. Doub et al. published a study in 2019 where they followed both male and female CML patients who had conceived during treatment with a TKI. In their study, 19.49% of the spouses of the CML male patients had a spontaneous abortion, which was similar to the rates in the general population. Hence, they concluded that male patients do not have to stop taking TKIs in preparation for conception [49].

#### 2.3.1. Imatinib

The US Food and Drug Association (FDA) does not put any contraindications for imatinib and has pregnancy as a precaution. They put imatinib as a class D drug for pregnancy, meaning that women should be advised to avoid pregnancy by using contraception when taking them as there is evidence of risk to the fetus. However, the benefits may outweigh risks and may warrant using the drug during pregnancy. In terms of male fertility, the FDA label mentions some of the animal studies that have had contradicting results on the effect of imatinib on male fertility. They mention that males who are worried about their fertility while on imatinib should consult their doctor, without putting it as a known adverse effect or contraindication to the medication, which is most likely due to the lack of information. Since imatinib is an inhibitor of tyrosine kinase BCR-ABL, c-Kit, PDGFR, and tyrosine kinase subclass 3 family, it is predictable that patients might experience signs and symptoms related to reduced androgen production like gynecomastia. Several studies confirmed this association as well as established associations between imatinib and increased progesterone concentrations. Chang et al. also found that imatinib crosses the blood–testis barrier, which reduces sperm density, count, survival, and overall activity. However, they concluded that imatinib had no effect on sex hormones or reproductive organs [39]. Another established effect of imatinib on children with CML is that it may alter and reduce growth velocity generally in adolescents and pre-pubertal children, especially with prolonged use [50,51].

Ault et al. published one of the first case series on conception among patients with CML treated with TKIs, specifically imatinib. They studied eight male patients who successfully conceived eight babies with a twin pregnancy and a single spontaneous abortion. One of the babies had a malrotation that resolved after surgical treatment. On follow-up after 38 months, no major issues were observed in the babies [52]. Some case reports have been reported of human cases of oligospermia due to the usage of imatinib prior to puberty [40]. However, conflicting data have been published on the effects of imatinib on sperm production, as some describe a reduction in sperm count related to imatinib and others find no effect on sperm production nor ability to conceive.

Imatinib has had promising results: Breccia et al. found that imatinib use was associated with successful conception during treatment in five cases. In these patients, sperm count showed no effects on sperm morphology and motility. The fetus development and growth were also normal [53]. Other published case studies have also shown similar results, in which prolonged high-dose imatinib therapy was used on CML patients [54]. This might suggest that even though imatinib may affect the PDGFR and c-Kit pathways, leading to reduced testosterone production, brief imatinib subjection may have minimal to no effect on male fertility as opposed to their female counterparts.

Animal studies have also shown that some tyrosine kinases needed for normal cellular functioning that are also inhibited by imatinib, like c-Abl, were found to play an important role in spermatogenesis. Hence, the c-Abl deficient mice were found to have impaired fertility. A case study conducted by Seshadri et al. described a patient who suffered from oligospermia. However, this patient was taking imatinib for hypereosinophilic syndrome rather than CML [33]. Hence, the authors suggested this risk must be discussed with patients prior to the initiation of therapy and that semen storage prior to treatment may be useful [33].

A study by Nurmio et al. on premature rats exposed to imatinib showed a delay in germ-line stem cell formation, as well as increased germ cell apoptosis and less spermatogonia proliferation. This raises the question of whether imatinib might show similar results in human children, since the process of testis development in rats and humans both rely on the same mechanisms [32]. In another animal study conducted by Prasad et al., it was found that imatinib does reduce intratesticular testosterone and increases LDH levels significantly in mice. Nevertheless, the researchers found that this is reversible once the drug is stopped [51]. The discrepancies seen in these studies might be attributed to the experimental designs of the studies.

#### 2.3.2. Nilotinib

The FDA also labels the effect of nilotinib on male and female fertility as unknown due to the limited number of studies. A recent study published in 2023 by Chethan et al. included thirty-eight CML male patients using TKIs. The authors observed that the patients went on to experience successful conception and healthy children. They also suggest that imatinib and nilotinib can be safely used in women during pregnancy [43].

A study involving mice conducted by Ozkavukcu et al. used nilotinib on male mice and found a significant reduction in the pregnancy rate in nilotinib-treated mice. They also studied the histological effects on the testicles of the mice and found lower spermatogenesis. In addition, they found that if either parent was on nilotinib, the fetus’s placenta was smaller and had abnormalities in the spongiotrophoblast and decidual zones. This confirms the effect of nilotinib on stromal proliferation of the placenta [36]. Another interesting finding was that first-generation mice born to a nilotinib-treated parent had lower chances of conception compared to controls [37].

#### 2.3.3. Dasatinib

The FDA label for dasatinib also does not highlight fertility as an established adverse event, but mentions that repeat-dose studies on different species have proven that the medication might potentially have a negative effect on fertility and reproduction. There are no restrictions for the use of dasatinib regarding conception in men. A study on human subjects using dasatinib was conducted by Cortes et al. in 2016, which included a small number of men who were treated with dasatinib and conceived. They reported good results with 91% normal outcome and only 6% spontaneous abortion, which is lower than the average rates. However, the small number of male cases in this study is a significant limitation. One of the infants had syndactyly, but they could not determine whether dasatinib was a factor [3]. Other case reports of men who conceived a healthy baby while on dasatinib therapy for CML have been published, showing no adverse effects [41,42].

An interesting recent study on mice by Garcia et al. where they used dasatinib and quercetin on mice to test their effects on testosterone and sperm demonstrated an increase in testosterone levels and sperm concentration, but did not affect sperm motility or fertility. Nevertheless, they still observed a reduction in seminal vesicle wight [35].

#### 2.3.4. Other TKIs

Other TKIs including ponatinib, bosutinib, and asciminib also have very limited information and studies on male infertility. The ponatinib FDA label mentions that it may impair male and female fertility, and fertility studies have not been yet conducted. Male fertility was not clearly highlighted in the FDA labels for bosutinib and asciminib. However, it is mentioned that asciminib may cause fertility issues in females. These drugs had no human studies and a few animal studies focusing on fertility. A 2020 study by Cortes et al. identified 33 successful pregnancies in patients who received bosutinib. Seventeen of those pregnancies were after paternal exposure to the drug, with nine live births, five abortions, and three unknown outcomes being the results of the pregnancies after paternal exposure. The authors concluded that there is no evidence of adverse effects on babies who are born to a father who was exposed to bosutinib at the time of conception; however, this still cannot be excluded due to the small number of cases available [38]. Novel agents like ponatinib and asciminib have no available data on their effects on male fertility [55]. However, ponatinib, a third-generation agent, has been studied on reproductive systems of rats and monkeys and showed degeneration of the epithelium of the testes [2].

### 2.4. Transplantation

Another CML treatment that may affect fertility is transplantation. Transplant patients are rarely able to conceive. However, this is rare in CML patients and is considered as a second- or third-line therapy if TKIs fail, which highlights the importance of discussing the potential effects that the disease and its treatments may have on fertility at the time of diagnosis [56,57]. In a 2022 study by Schleicher et al., patients with pediatric CML who were treated with hematopoietic stem cell transplantation were followed and answered a questionnaire about several complications including infertility, which was reported in 24% of patients. One male patient reported reduced testicular volume that was diagnosed after the transplantation. However, most patients reporting infertility were females. The male patients indicated hypogonadotropic hypogonadism after irradiation. Nevertheless, this study has a limitation in that it depends on a questionnaire that relies on the self-identification of infertility [58].

## 3. Limitations

As previously mentioned in this review, the main limitation is the lack of systematic studies that investigate the effect of CML medications on fertility in humans, especially in males. In addition, there is a huge disparity in the information taken from animal studies when compared to the conclusions made in human male studies. Further, the few human studies that are available have different conclusions and demonstrate discrepancies in their results. This might be attributed to the different study designs and doses used when applying these drugs in animal models. Some human studies were also dependent on subjective results from questionnaires and self-reporting, which might yield inaccurate or flawed results. It can also be explained by the variety of animal models used in studies and the difference in their response to the aforementioned drugs. Hence, more studies on this subject would improve the current understanding and help with the contradicting evidence. Table 2 summarizes the main human and animal studies included and their limitations. It is clear that more prospective studies with greater sample sizes are necessary to deepen our understanding and to obtain sufficient evidence to start discussing changes in the current guidelines and usage of these therapeutic approaches.

## 4. Conclusions

This review highlights the necessity of well-controlled studies with larger numbers of subjects and longer follow-up periods to improve our understanding of the short- and long-term effects of CML and its treatments on male fertility and sexual functioning. According to the current evidence, it would not be advised to blindly discontinue CML medication in male patients who desire to conceive, especially in more established drugs with a relatively known profile like imatinib. Nevertheless, the subject of fertility should be a point of discussion when choosing the most appropriate drug and when explaining the possible adverse events to patients, especially in newer medications with fewer available studies. This becomes more necessary considering the current evidence of the rise in CML diagnoses in young adults in developing countries. It is also necessary to carry out more molecular studies that identify the particular stages of spermatogenesis affected by each drug therapy in order to help provide novel treatment strategies. Male patients should be advised to consult their hematologist if they try to conceive while they are being treated or when experiencing active disease. The additional possible effects like the presence of congenital defects, impaired growth, or malignancies in subsequent generations also need more research, and the discussed associations should be disclosed with patients.

## Figures and Tables

**Figure 1 cancers-16-00791-f001:**
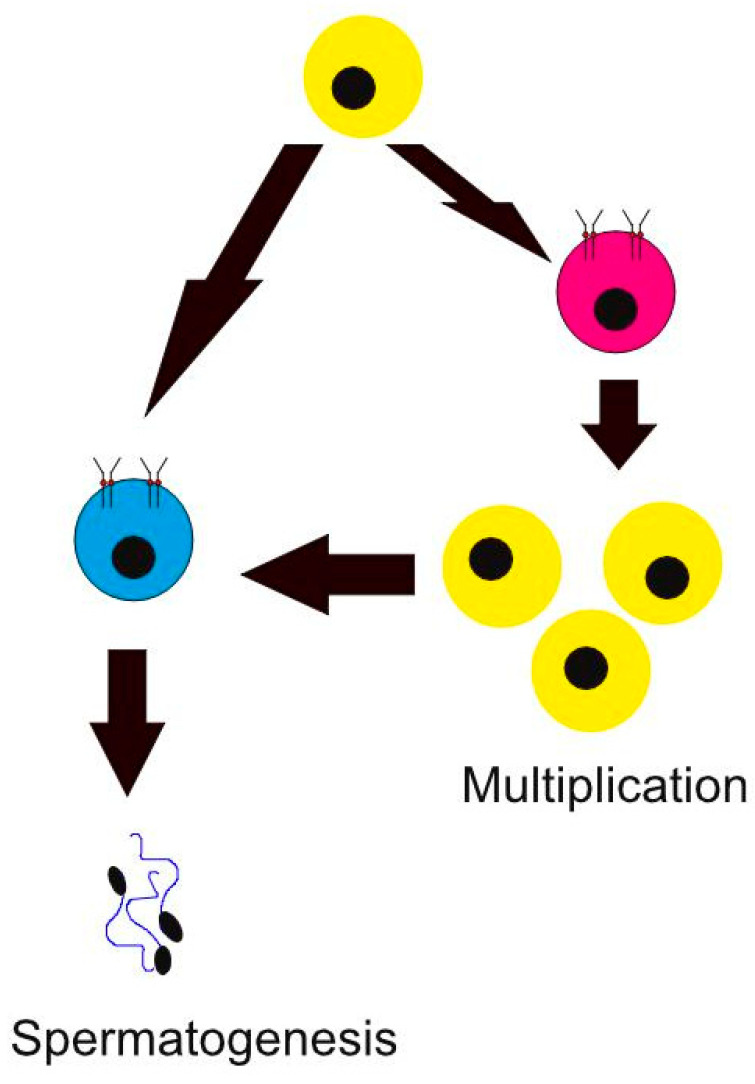
Kit-negative (Kit−) spermatogonial stem cells (yellow) have the ability to either transform into Kit-positive (Kit+) spermatogonial stem cells (pink) or Kit-positive (Kit+) differentiating spermatogonia (blue). Kit+ differentiating spermatogonia can produce sperm cells to complete spermatogenesis. Kit+ spermatogonial stem cells can help increase the number of available Kit- spermatogonial stem cells through multiplication.

**Table 1 cancers-16-00791-t001:** Summary of the data available on TKIs and fertility.

Medication	Imatinib	Dasatinib	Nilotinib	Ponatinib	Bosutinib	Asciminib
**FDA information on fertility**	Not highlighted—males who are worried about their fertility while on imatinib should consult their doctor.	Not highlighted in males.	Not highlighted in males.	May impair male and female fertility.	Not highlighted.	Not highlighted.
**Animal studies**	Available. Studied on mice and rats: studies show impaired fertility and negative effect on germ-line stem cell formation [32,33]. Was found to reduce intratesticular testosterone and increase LDH levels significantly, but was reversible [34].	Available. A study on mice showed no effect on fertility, but a reduction in seminal vesicle weight [35].	Available. Study on mice found lower conception rates in nilotinib-treated male mice. Offspring had lower chances of conception [36,37].	Available. Studies on rats and monkeys showed degeneration of epithelium of the testes [2].	Available. Bosutinib resulted in reduced fertility in males as demonstrated by 16% reduction in number of pregnancies. It was not mutagenic in a battery of testes [38].	Not available.
**Human studies**	Available. Several case studies show no effect on male fertility. Studies also show no effect on sperm morphology and motility. Other reports show effects like reduction in sperm density, count, survival, and overall activity [39,40].	Available. A 2016 study showed no effect on conception; however, the study had a small sample size as a significant limitation. Several case studies of normal conception while taking medication [3,37,41,42].	Available. A study with 38 males showed it can be used safely in both men and women [43].	Not available.	Not available.	Not available.

**Table 2 cancers-16-00791-t002:** Summary of the major animal and human studies that study the effect of CML drugs on male fertility.

Paper	Type of Study	Conclusions	Drug Used	Limitations
[20]	Animal study—transgenic sickle cell mice	Hydroxyurea use was associated with reduction in testosterone levels and sperm motility (negatively affects fertility). Leydig cell hyperplasia was observed.	Hydroxyurea	Animal study with sickle cell models, which may not be an indicator of the effects seen in CML patients. High doses of nitric oxide could be the main cause of the adverse effects.
[21]	Human study—retrospective review including male adult patients with sickle cell disease	Azoospermia observed. Effects are reversible after discontinuation.	Hydroxyurea	A case-series. External validity compromised. Study conducted retrospectively. No baseline data recorded before hydroxyurea usage.
[32]	Animal study—rats	Imatininb was associated with impaired fertility in rats as it interferes with maturation processes in rat testes. This includes gonocyte migration, testis growth, formation of spermatogonial stem cell and Leydig cell pools, and proliferation of type A spermatogonia.	Imatinib	Animal study on rats. Further human studies are needed to show the observed effect in human testis.
[34]	Animal study—Swiss albino mice were and their testes were studied after imatininb exposure	There is an effect on LDH and testosterone exerted by imatinib, but it is reversible.	Imatinib	Animal study on mice. Further human studies are needed to show the observed effect in human testis.
[39]	Human study—the effects of imatinib were observed on sperm parameters by a computer-assisted sperm assay and the male reproductive system by using ultrasound	Imatinib crosses the blood–testis barrier and has adverse effects on sperm production.	Imatinib	Limited sample size.
[35]	Animal study—mice	Dasatinib showed no adverse effect on fertility.	Dasatinib	Animal study. Human studies are needed to show the observed effect in human testis. This study also included quercitin with dasatinib as senolytic agents. A study examining the effect of the drug on CML subjects is needed.
[3]	Human study	No effect on conception in cases subjected to dasatinib.	Dasatinib	Small sample size was a significant limitation.
[38]	Human study	No evidence of adverse effects on babies born to a father who was exposed to bosutinib at the time of conception.	Bosutinib	Limited sample size.

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
