# Peer review of "Male Fertility and Fatherhood in Chronic Myeloid Leukemia: Current Understanding and Future Perspectives"

_cancers, 2024, doi:10.3390/cancers16040791_

Round 1
Reviewer 1 Report
Comments and Suggestions for Authors
The review article is comprehensive and very relevant to the field of male infertility. It is also shed the light on a good recommendation for physicians to advice young patients regarding the possible effect of the different types of treatment on their future fertility. It is important to consider fertility preservation when providing the optimal treatment.
However, the review is missing the mechanisms of action and how the different medications adverse male fertility; including the stages of spermatogenesis (pre-meiotic, meiotic and post-meiotic) that may be affected. This is important because it may also provide possible future therapeutic male infertility strategies.
Author Response
Dear Reviewer 1,
Thank you for your kind words about the impact and comprehensiveness of this research paper. Please find all pointers addressed below
However, the review is missing the mechanisms of action and how the different medications adverse male fertility; including the stages of spermatogenesis (pre-meiotic, meiotic and post-meiotic) that may be affected. This is important because it may also provide possible future therapeutic male infertility strategies.
Thank you for this comment. Some more insight about the mechanism in which these therapies may affect male fertility has been added. Due to the lack of studies that specify the stage affected, we also added it in our recommendations. The diagram used and explanations regarding molecular effects provide details about the direct effect of these drugs on certain stages in sperm production.
Reviewer 2 Report
Comments and Suggestions for Authors
Please revise the title to: "Male Fertility and Fatherhood in Chronic Myeloid Leukemia: Current Understanding and Future Perspective"
Figure 1 is not informative please add more details on diagram.
Table 1 needs to be updated. for example this article is available for Asciminib : "Asciminib Mitigates DNA Damage Stress Signaling Induced by Cyclophosphamide in the Ovary"
Most of articles in Discussion section are old references and needs to be updated.
The result of search strategy is not clear and its better to demonstrate on a Prisma graph.
The article suffers greatly from the lack of diagrams or tables that can replace the text and better show the goals of the article and reveal the results. Therefore, many places need to design a table or diagram.
Author Response
Dear Reviewer 2,
Thank you for your comments. We are sure that these have helped us improve the paper. Please find all comments addressed below.
Please revise the title to: "Male Fertility and Fatherhood in Chronic Myeloid Leukemia: Current Understanding and Future Perspective"
Thank you for your comment. The title has been modified as suggested.
Figure 1 is not informative please add more details on diagram.
Thank you for your comment. More details about the process has been added in the text.
Table 1 needs to be updated. for example this article is available for Asciminib : "Asciminib Mitigates DNA Damage Stress Signaling Induced by Cyclophosphamide in the Ovary"
Thank you for your comment. The study referred to was looked into thoroughly, it unfortunately only holds information about the effect of Asciminib on female fertility only. The table summarizes current evidence about effects on male fertility. The table was updated, and more studies were added as references. One of the aims of this paper is to highlight the lack of evidence and studies done to study male fertility in some of the novel TKIs used to treat CML in young adults.
Most of articles in Discussion section are old references and needs to be updated. The result of search strategy is not clear and its better to demonstrate on a Prisma graph.
Thank you for your comment. Since this is a literature review rather than systematic review, we understand the confusion that might have been caused by the “methods” part. Hence, this part was removed to keep in line with a literature review provided by our paper.
The article suffers greatly from the lack of diagrams or tables that can replace the text and better show the goals of the article and reveal the results. Therefore, many places need to design a table or diagram.
Thank you for your comment. A new table has been inserted in the paper summarizing all major studies and their limitations. They also show the goals and reveal conclusions as per your kind suggestion.
Reviewer 3 Report
Comments and Suggestions for Authors
It is my pleasure to review this paper entitled “Male Fertility and Fatherhood in CML: Current Understanding and Future Perspective” This review aims to add male fertility to the equation to facilitate therapy individualization for younger male patients seeking to preserve their fertility during or after CML therapy, when applicable. The topic is quite interesting. However, there are some drawbacks that could be addressed before an eventual publication. Introduction should end with the aim of the paper and in this case, it is too much long. Please revise them.
Please add a figure with the search strategy
Please add inclusion and exclusion criteria
Authors should highlight the strengths and limitations of the paper.
Author Response
Dear Reviewer 3,
Thank you for your kind words about our paper. Please find all comments addressed below.
However, there are some drawbacks that could be addressed before an eventual publication. Introduction should end with the aim of the paper and in this case, it is too much long. Please revise them.
Thank you for your comment. The introduction has been modified to end with the aim of the paper. The introduction has also been trimmed and more details has been provided in the following section instead to shorten the introduction as suggested.
Please add a figure with the search strategy
Please add inclusion and exclusion criteria
Authors should highlight the strengths and limitations of the paper.
Thank you for your comment. To stick with the essence of this paper being a comprehensive literature review that may help clinicians in their consultations with CML patients starting one of the therapeutic strategies discussed, we decided to remove the methods part to avoid confusion.
As per your suggestion, a new table was added that discusses the aim of the included and reviewed papers as well as their limitations.
Round 2
Reviewer 2 Report
Comments and Suggestions for Authors
I do not have more suggestion.
Reviewer 3 Report
Comments and Suggestions for Authors
I want to thank you tha authors for the corrections